# Online University Counselling Services and Psychological Problems among Italian Students in Lockdown Due to Covid-19

**DOI:** 10.3390/healthcare8040440

**Published:** 2020-10-29

**Authors:** Giulia Savarese, Luigi Curcio, Daniela D’Elia, Oreste Fasano, Nadia Pecoraro

**Affiliations:** Centre of Psychological Counselling, Campus di Baronissi, University of Salerno, 84081 Baronissi (Sa), Italy; lcurcio@unisa.it (L.C.); ddelia@unisa.it (D.D.); ofasano@unisa.it (O.F.); npecoraro@unisa.it (N.P.)

**Keywords:** Covid-19, university students, Psychological Counseling, anxiety and depression, psychosomatic symptoms, PTSD, online learning difficulties, quality of life

## Abstract

*Introduction:* With the advent of Covid-19, Italian university students were overwhelmed by fear of the pandemic and the social restrictions of the lockdown phase, with all didactic activity provided online. These stress factors caused people to experience psychological problems and/or the aggravation of pre-existing mental symptomatology. Psychological support is, therefore, important for the university-student population. *Aims:* (1) Analyzing the psychological difficulties and mental problems relative to lockdown from Covid-19 of students who asked for help from the Center of Psychological Counseling of the University of Salerno. (2) Describing the online services of the university’s psychological counseling treatment. *Participants:* 266 university students, but only 49 were undergoing psychological treatment during the Covid-19 lockdown at the center. *Methods:* (1) Semistructured interview; (2) Questionnaire consisting of sociodemographic information and ad hoc questions; and (3) Scl-90-r test. *Results and Conclusions:* Aim 1: The main results highlight high levels of anxiety and stress, concentration disorders, and psychosomatization. In several cases, there was a reactivation of previous traumas and sleep was found qualitatively compromised. Aim 2: Counseling services included telephone listening activities, online psychological interviews, psychoeducational groups for interventions of anxiety management, and workshops on study methods conducted in small groups. The online counseling intervention, in times of emergency, increased the resilience and identified any psychological problems in order to implement timely management.

## 1. Introduction

The SARS-CoV-2 or Covid-19 coronavirus emerged at the end of 2019 in Wuhan, China. In a few weeks, it turned into a global health emergency. Italy was among the countries with the highest number of cases. The Italian lockdown, so-called Phase 1, lasted from 10 March to 3 May 2020. In Italy, as in the whole world, the Covid-19 emergency has caused isolation, social distancing, the abrupt interruption of direct interpersonal relationships, the loss of work, disorientation, uncertainty about future times, and economic hardship, leading many people to experience anxiety-like conditions.

These are just some of the main stress factors that can normally contribute to the onset or aggravation of pre-existing anxious symptomatology [1,2,3,4].

The experienced situation led part of the population towards the emergence of previous but silent pathological conditions or the worsening of patients already suffering from anxiety, stress disorders, depression, and other problems related to mental balance [1].

The forced and sudden interruption of daily habits, the prolonged period of isolation from social life, the distance from places of study or work, the forced cohabitation with family without moments for themselves, or total isolation in solitude, constituted sources of stress and often strained the general population, particularly the students’ ability to cope positively with obstacles and tolerate frustration while waiting to see their needs satisfied with autonomy, relational and connected to the continuation of their studies [1,2].

To add to this delicate and complex situation, anxiety is generated by an invisible enemy, the Covid-19 virus, new to medicine and violent in its outcomes, which entails constant fear of being infected or infecting loved ones, and concerns about the future and uncertain life prospects.

### 1.1. Studies on Psychological Difficulties of College Students during Covid-19 Lockdown

Many studies addressed the risk factors and prevalence of symptoms [3,4] of acute post-traumatic stress disorder (PTSD) in university-student populations. 

In China, Liu et al. [5] explored university students’ cognition, psychological status, anxiety, and depression, highlighting that there were statistical differences between the different genders of students in the manifestations of distress and the expression of emotional panic, and in exposure to the risk of contagion. In addition, the levels of anxiety and depression of university students in China during the Covid-19 pandemic were higher than those in the normative data of the national population. The discomfort of exposure to the risk of contagion was a risk factor for depression. Cao et al. [6] reached the same conclusions, indicating that 0.9% of respondents had severe anxiety, 2.7% moderate anxiety, and 21.3% mild anxiety. Furthermore, living in urban areas, having a stable family income, and living with parents were protective factors against anxiety. Lastly, having relatives or acquaintances infected with Covid-19 was a risk factor for an increase in anxiety disorders in college students. Social support was negatively correlated with the level of anxiety. 

In Spain, Odriozola-González et al. [7] analysed the psychological impact on the university community during the first weeks of Covid-19. A cross-sectional study was conducted. The depression, anxiety, and stress scale (DASS-21) showed moderate to extremely severe levels. Many respondents had experienced moderate-to-severe anxiety compared to that in the beginning of the pandemic. Students from the humanities, and social and legal sciences showed higher scores in relation to anxiety, depression, and stress disorders than those of engineering and architecture students. 

In Kosovo, Arënliu et al. [8] identified a trend in the interviewed university students: during the lockdown, those who spent more time on social media reported more severe or serious psychological distress. A similar trend was also found in students who showed little motivation for online studies.

In Pakistan, Salman et al. [9] found moderately severe anxiety and depression scores in students. Respondents aged ≥31 years had significantly lower depression scores than those younger than 30 did. Males had significantly lower anxiety and depression scores than females did. In addition, those who had a family member, friend, or acquaintance infected with the disease had a significantly higher anxiety score. The main sources of suffering were related to adverse effects of the ongoing pandemic on daily life, followed by the rapid spread of the disease. As far as coping strategies are concerned, it was discovered that most of the respondents had adopted “religious/spiritual” coping.

In Jordan, Ala’a et al. [3] noted that most of the surveyed students were considered to be afflicted with severe psychological distress. Females had a statistically significant higher percentage of mild and severe psychological distress. Many students reported that they were not motivated for distance learning. There was a statistically significant inverse relationship between severe psychological distress and motivation for distance learning.

The most common coping strategy among students was spending more time on social media. In addition, a few of the students reported using drugs to address Covid-19-related discomfort, and many students reported that distance learning was their most serious concern.

In India, Duan and Zhu [10] studied the lives of university students during the time of Covid-19, highlighting that they were disrupted in their overall structure as rapid changes had occurred in their lifestyles, particularly in terms of study, work, and social gatherings. So many uncertainties arose in them on academic and career issues, on getting sick with Covid-19, on the duration of the pandemic, and on what the near future holds. The aftermath of this was manifestations of anxiety, worry, fear, frustration, insecurity, depression, and despair. The authors noted that this could depress students, and influence their well-being and academic success. The authors therefore suggested that psychological-support interventions focus mainly on eliminating a pessimistic view of life, and favoring an optimistic and resilient approach. According to the authors, it is necessary to work on enhancing coping skills.

In Italy, Gallè et al. [11] conducted a “survey on the knowledge and behavior of university students during the Coronavirus-19 Pandemic” (EPICO study), which took place in the last two weeks of March 2020, involving three Italian universities: La Sapienza, University of Rome; Parthenope, University of Naples; and Aldo Moro, University of Bari. The entire estimated population was 166,703 students. This survey aimed to understand university students’ ideas about national training and employment measures, including the closure of schools, universities, and workplaces after the lockdown. In addition, the authors wanted to explore students’ level of knowledge about the Covid-19 pandemic, and the adopted behaviors during the blockade. Results showed a good level of knowledge of the Covid-19 pandemic and its control, mainly among students attending life-science courses. Most students did not change their diet and smoking habits, while most of the sample reported reduced physical activity. The authors also concluded that the confinement gave students more opportunities to watch television and surf the net while staying at home. In this way, they were able to improve their level of knowledge on the evolution of the pandemic. Mass media had a significant influence on both people’s knowledge and attitudes, and on their perception of risk. Regarding the examined lifestyles, ministerial recommendations were generally followed. 

### 1.2. Studies on University Psychological Support during Covid-19 Lockdown

With regards to university psychological-counseling experiences supporting the Covid-19 experience, Duan and Zhu [10] reported that the National Health Commission had published guidelines for emergency interventions on psychological crisis for people affected by Covid-19. These guidelines highlight the need for timely healthcare mentality from the early stages of the pandemic, and how, in reality, patients with suspected infection, family members in quarantine, and medical personnel were poorly managed in China. Even interventions on university students should have been based on a global assessment of the risk factors that more easily lead to the onset of significant psychological problems in the event of an emergency. These factors include bad mental health before a crisis, mourning, self- and hetero-directed aggressive behavior with injury to themselves and/or family members, life-threatening circumstances, panic, separation from the family, and low family income. 

Zhai and Du [12] reiterated some actions of the universities that were considered essential at the time of Covid-19. First, in addition to distance education, student advice should have continued, passed on by telecommunication (e.g., telephone calls, online advice) in order to provide academic support to students. All university staff should have considered offering virtual office hours to students, and worked together to keep connected, and help students to process and address the academic concerns caused by the interruption of the semester. Second, for students whose traineeships or research projects were affected by the pandemic, the trainee site supervisors and research consultants should have actively engaged in seeking alternative plans, allowing them to work from home to maximize internship and research experiences. Third, universities were expected to work on innovative methods to support students in writing dissertations. University counseling should have to set up all options to continue to provide students with remote consulting. In addition, university counseling could have provided students with options to join online support groups that would allow for them to share common concerns and receive social support. In addition, they could have made joint projects with other bodies to develop containment strategies and transmit public-health messages to students, urging positive coping resources, and encouraging them to take action to protect their mental health.

Novo et al. [13] identified psychological well-being in college students as an indicator of academic success, skill development, and personal growth. The results of their study show a significant relationship between academic involvement, and psychological and occupational well-being. In addition, the assessed students during Covid-19 showed worse psychological well-being, less satisfaction with their results, and less social participation. These figures were considered fundamental by the authors to guide the design of counseling-psychology programs to help improve the well-being and academic success of students.

In the Philippines, Toquero [14], using data updated to 6 April 2020 by UNESCO, reported that ways of teaching and learning had changed substantially due to Covid-19, and that this had affected 1,576,021,818 students around the world, or 91 (33%) of the students enrolled in a total of 188 countries. However, there are still few studies on the experiences of consultancy centers in universities. Zhai and Du [12] were of the same opinion upon analyzing the Chinese context.

Liy et al. [15], and Sahu [16] focused on the psychological impact of the closure of universities due to Covid-19. He concluded that students were concerned about widespread fears that the Covid-19 pandemic would adversely affect exam performance, and the need for universities to support students’ mental health by providing assistance, online lessons, and strategies to manage pandemic stress. The study also highlighted how universities should have paid more attention and provided more systematic support to vulnerable international students.

## 2. Aims

Starting from analysis of the existing literature [10,12,13,14,15,16,17,18] and the consideration that we did not find studies on concrete experiences of support to university students during the lockdown, we aimed to:(1)Analyze the psychological difficulties and mental problems to lockdown from Covid-19 of students who asked for help from the Center of Psychological Counseling of the University of Salerno; and(2)Describe the online services of the University of Salerno psychological-counseling treatment.

## 3. Methods

### 3.1. Participants

The recruited participants were 266 university students, but of these, only 49 were undergoing psychological treatment during the Covid-19 lockdown at the Center of Psychological Counseling of the University of Salerno.

They requested psychological support in the Italian lockdown phase between 12 March and 3 May 2020.

Participants were between 20 and 35 years old (M age 24.89; DS 3.5). They had been enrolled in the first to sixth years of their respective courses (M enrolment year 2.6; DS 1.5) and participated in 1 to 6 counseling sessions (M sessions 3; DS 1.3).

Twenty-six were females and 23 males; 48% lived in university residences because they lived far from home, and 52% lived with their family in their country of residence; 10% had come from foreign countries and studied in Italy as guests of Salerno university residences.

This is a preliminary study and for observational purposes; it was not possible to have a comparison group. Follow-ups are ongoing.


*Data Collection and Procedures*


−A semistructured interview consisting of 10 basic questions was conducted on the first meeting of the analysis interview, which assessed the general mental state in line with indications of the Order of Italian Psychologists. It used questions to assess different domains of mental functioning: speech, emotional expressiveness, thinking and perception, and cognitive function. A semistructured interview was constructed ad hoc and included a checklist by the psychologist. Scores were dichotomous: (1) no mental-functioning difficulties in the investigated domains; (2) mental functioning difficulties in the investigated domains.−A questionnaire consisting of (i) sociodemographic information and (ii) 19 ad hoc questions (based on existing questionnaire of the Order of Italian Psychologists) in five areas: (1) psychological manifestations, (2) psychosomatization, (3) family and social relations, (4) distance learning/study difficulties and related matters, and (5) quality-of-life indicators. −The Scl-90-r test [19], which investigated the presence of symptoms detected during the counseling course. The measurement properties of Scl-90-r were standardized factor scores at admission, these being 0.04 higher than at discharge and 0.06 higher than those of controls; the ability to identify those with and without a psychiatric disorder was good (area under the curve (AUC) = 83%; Glass’s Δ = 1.4; Cohen’s *d* = 1.1; diagnostic odds ratio, 12.5). Scores were also fairly sensitive to change between admission and discharge (AUC = 72%; Cohen’s *d* = 0.8).

After administration of the semistructured interview, questionnaire, and test, the students had a psychological-counseling interview with a maximum of six meetings on a weekly basis. The psychotherapists conducted the interviews using an online digital platform. Students could also follow psychoeducational online groups for interventions of anxiety management and an online workshop on study methods, with activities conducted in small groups. 

For data analysis, we used the summarised descriptive statistics and statistical package SPSS-26.

### 3.2. Ethical Considerations

Information on the study protocol was given, and informed consent was obtained before administration of the questionnaire at the first online appointment at the Center of Psychological Counselling of the University of Salerno.

The study was conducted according to the guidelines of the Declaration of Helsinki. The study was absent of risk or burden, sponsors, conflicts of interest, and incentives for the responding subjects.

The study was conducted in accordance with the legislation of the Italian Code regarding the protection of personal data (Legislative Decree n. 196/2003). Participants were informed about the general purpose of the research, the anonymity of the answers, and the nature of voluntary participation and signed informed consent. No incentives were given. 

The research complied with the Ethics Code of the Italian Psychology Association (Associazione Italiana di Psicologia—AIP, 2015) that draws inspiration from the WMA-Declaration of Helsinki (1964/2013). As no Institutional Review Board for Psychology research was available from the affiliations of the researchers involved in the study (University of Salerno), no request for approval could be submitted.

## 4. Results 

We summarize the main results of the 217 students who did not follow the psychological-support path at the psychological-counseling center. Of the subjects, 44% often felt anxiety; 48% experienced psychological and physical discomfort; 56% admitted that they only rarely felt really well; 58% felt tired; and 71% admitted that they often experienced fatigue in study activities.

We only focus in detail on the results of the 49 students who used the services and psychological treatment provided by the Psychological Counseling Center.

In the semistructured interview, all subjects scored 1 at all domains, so there were no “mental-functioning difficulties”.

Results of the students’ responses to the questionnaire are shown below. Five areas were identified. 

The first area concerned psychological manifestations that prompted students to request help from the Please note that author names, affiliations and e-mail could not be changed if paper accepted, so please check it carefully when revising your manuscript.Center of Psychological Counseling (Table 1).

All students declared having experienced anxiety disorders (yes = 100%, no = 0; Table 1.1). The symptomatology connected to this condition included typical manifestations of generalized anxiety, such as restlessness or constant psychic tension, feeling drained, chronic fatigue, difficulty concentrating with a consequent reduction in memory, nervousness and irritability, muscle tension, sweating, and difficulty sleeping (defined as difficulty in falling asleep and maintaining sleep or restless sleep). Some students had also experienced panic attacks and the sense of a threat perceived in an invasive way. These were connected to the reactivation of past traumas.

In addition to manifestations of anxiety, 87% of the students said that they had experienced distressing symptoms attributable to a depressive mood disorder (Table 1.2), such as a decrease in interest in things, a sense of tiredness, disorientation, apathy, difficulty concentrating, altered sleep patterns, excessive fear, and fear for present and future prospects. The students stated in the interviews that depressive experiences had often been connected to the sense of powerlessness regarding the unfolding of events, and the sense of lost relationships. Therefore, rather than activating coping strategies, they closed in on themselves in an avoidant or withdrawn mode.

Rumination, as a cognitive component of anxiety and depression, even on the obsessive side, presented itself with negative intrusive thoughts for the present and the future, accompanied by negative images (for example, “while I complete an online examination, I have problems with the lines”) and negative feelings (for example, “I see the future as a black cloud”); 77.6% of students had such thoughts (Table 1.3). In some cases, exams rather than didactic objectives became the primary nucleus of concern around which the thoughts rotated, unable to find other ways due to the forced closure and the narrowing of social relationships, and the impossibility of identifying other modes of expression. In general, there was a reduction in cognitive and emotional functioning compared to some areas at that time perceived as founding existence.

Anxiety conditions, sometimes blocking daily study activities, the sense of disabling sadness, and the presence of persistent dark thoughts of an anxious or distressing nature towards the future, resulted in a feeling of fragility of the self in the students, affecting their sense of self-esteem, and above all their self-efficacy as a perception of efficiency in daily performance (Table 1.4); this was especially true in subjects who had already not had a high level of self-esteem prior to the pandemic.

Specifically, with regard to the observed post-traumatic symptoms (intrusiveness, avoidance, hyperarousal, cognitive and emotional alterations, dissociative symptoms), we distinguished the one attributable, in purely nosographic terms, to a PTSD (Table 1.5) and compatible with acute stress disorder (Table 1.6), where the stressful event was considered Covid-19 itself.

During anamnestic analysis, students with probable PTSD were identified, namely, those who had reported traumatic events related to previous experiences such as bereavement, abuse, illness and health interventions, and bullying. These are experiences that expose someone or a loved one to “real death or death threat” (events that saturate DSM-5 Criterion A for the diagnosis of PTSD).

From the anamnestic collection, it emerged that, for most of these subjects, the most recurrent post-traumatic functioning was the “persistent avoidance of stimuli associated with the traumatic event, starting after the event itself” (Criterion C of the DSM-5), that is, they avoided, persistently and even for years, unpleasant memories, thoughts, and feelings associated with the event. The lockdown countered this mechanism, both in forcing people to stay at home (who were no longer able to avoid memories linked to that place, for example) and by conveying news of grief and danger, thus reactivating traumatic memories. Lastly, news spread by the media regarding the death and loss of family members probably took the form of triggers, that is, they reactivated previous traumas in some students that had remained not completely processed.

In this regard, 24.4% of the students reported the reactivation of a trauma (Table 1.5), expressing a discomfort deriving from perceiving their current mental state as similar to that experienced in a stressful moment in the past. The psychopathological outcomes were PTSD and dissociation. In these cases, in addition to counseling work focused on trauma, were the needs of psychiatric paths and psychotherapeutic interventions in mental-health services in Phase 2 after the lockdown.

On the other hand, the pandemic, which could still be a highly stressful event (from Criterion A), did not in itself lead to post-traumatic symptoms linked to Covid-19 (0% of the interviewees declared to suffer from acute stress; Table 1.6). None of the students interviewed reported being directly affected by Covid-19 or having family members contaminated with the virus.

With regard to the second identified area, psychosomatization, 79.6% (Table 2) of the students reported disorders involving the body, especially at the gastric and intestinal level, but also muscle tension and headaches. As considered previously, in many cases, these manifestations represent a failed, partial, or inadequate elaboration of anxiety-provoking and stressful experiences of a person. 

The third area concerns students’ family and social relationships (Table 3), as perceived and experienced in the first Covid-19 phase.

Of the students, 53.1% said that forced cohabitation was one of the main causes of stress, so much so that some of them preferred to live in the university residences in the quarantine phase rather than returning to their families. In many cases, especially for those away from home, returning to the family required a readjustment to the rhythms and rules to which they were no longer accustomed (e.g., mealtimes). It also limited gained autonomy, forcing them to redefine relationships with loved ones in relational terms. In some cases, coexistence was difficult because it was sudden and forced, characterized by unease due to a lack of privacy or because of spaces that were not always adequate for study. However, 46.9% considered this coexistence an opportunity to cultivate time with their family members (Table 3.1). The presence of complicated or difficult relationships with a family member or between family members was one of the major risk factors for stress generated by forced cohabitation.

Difficulties in long-distance relationships, the redefinition of friendships, or separation from a partner represented major stressors for the students. In this regard, 83.7% of the students said that they had experienced these difficulties, while 16.3% were not affected by the lockdown in this regard (Table 3.2). Some students ended romantic relationships or friendships.

The inability to cultivate relationships in person or participate in social routines, such as meeting friends on Saturdays or enjoying social hobbies, generated a state of unease in 69.4% of students (Table 3.3).

In this age group of identity transition, relationships with peers certainly represent a fundamental fulcrum of existence for most students, both for the function of sharing, and for evolutionary thrust and elaboration.

In general, good family and social relationships were a protective factor regarding anxiety. 

The fourth area concerns distance teaching, study, and related difficulties.

Of the students, 51% said that they had experienced difficulties associated with distance learning (Table 4.1) mainly due to problems related to access and sometimes a lack of suitable equipment. Other issues were the disruption of everyday university life, with a lack of opportunities to foster relationships with teachers and colleagues; tiredness in trying to follow online lessons not always properly organized on the screen; and the continuous reorganization of exam modalities. For many students, distance examinations were an enormous source of stress due to the presence of other students when they were viewing their exam, and the exposure of their work spaces to family members, sometimes experienced as a violation of privacy.

Of the students, 49% (Table 4.2) said that the reorganization of their study time, their study spaces often being shared with the family, and the need to review their study methods in line with online teaching were sources of stress that affected concentration.

For 51% of students, online teaching did not create difficulties regarding these indicators. Being able to commit time to studying and lessons was a protective factor regarding lockdown stress, making students rediscover in adaptive terms that they were capable of completing tasks in a pre-established period of time, feeling effective in their learning actions.

The relationship with time was more complex, as it was not marked by routine activities outside one’s home. Of the students, 51% had difficulty organizing their time (Table 4.3), perceiving it as dilated, while 49% thought that there was no marked difference with the student condition experienced before the lockdown, which partly coincided with the home-studying phase of exam preparation. “Suspended time” waiting for news on the evolution of the pandemic, if not filled with activities aimed at concrete objectives, was a source of anxiety and tension for some of the interviewed students. 

The fifth area concerns some indicators of quality of life: perception of feeling fit, quality of sleep, food intake, physical activity, and smoking and alcohol intake (Table 5).

Of the students, 65.3% declared that they did not feel fit (Table 5.1), and 73.5% said that they did not sleep well (Table 5.2). Students’ appetites were less affected, with 42.9% of students saying that they had experienced appetite alteration (Table 5.3), and 81.6% of students did not practice any form of sport at home (Table 5.4). Only 18.4% said that they had smoked more (Table 5.5), and 4.1% said that they had consumed more alcohol (Table 5.6).

In other words, students felt less fit and presented sleep difficulties, in line with their emotional experiences. 

Five students were sent to territorial mental-health units for psychiatric consultations, as the presence of severe clinical symptoms was verified by psychotherapists through interviews (noting behaviors attributable to psychopathological disorders and risk factors for remote or family symptomatic anamnesis) and through the administration of Test Scl-90 r.

## 5. Online Psychological Counseling: Description of Services and Treatment

At the Center of Psychological Counseling of the University of Salerno [20,21,22,23,24], offered services were reorganized to meet the new and changing needs of students who had requested psychological help. The reorganization concerned both the type of offered services, and operational and management methods, according to an emergency type of intervention.

Online clinical reception, listening, and treatment services were activated for students who reported a need for psychological attention. The students contacted the Center by sending an email to counselling@unisa.it, which is the normal channel of contact. A phone number was added to this for use every day at specific times to manage the emergencies, and to provide a listening channel and more direct support. 

If necessary, the students were sent to territorial mental-health services.

The professional references for operating online were first the “Managing stress during the Coronavirus pandemic “guidelines published by the World Health Organization (WHO) [25]. This guide presents a series of practical tips to be implemented daily in crisis management, and illustrates how “during a crisis, it is normal to feel sad, stressed, confused, frightened, or angry”. It highlights that, through dialogue with experts, it is possible to clarify and manage these moods. Furthermore, in order to deal with the emergency with knowledge, reference was made to guidelines developed by the National Council of the Order of Italian Psychologists [26]. Starting with important considerations regarding psychological intervention mediated by the web [27], these guidelines illustrate, among other things, how to rationalize the fear of coronavirus, and some good practices to use in this period [26,27].

The psychological interviews were carried out from Monday to Friday, respecting privacy and following the signing of informed consent through a blended online platform that took into account privacy needs. The counseling path predicted four meetings on average, with weekly frequency and a meeting duration of about 50 min. The use of the online mode significantly changed the professional work of psychotherapists. It was necessary to carry out daily team meetings to promptly and safely manage all emerging issues, including the modification of the setting and the professional reflections that this activated (in line with Order of Italian Psychologists—CNOP-guidelines) [28,29].

In addition to the counseling process, small groups were organized on the online platform in response to the specific needs of users: one for the psychoeducational management of anxiety and the other on study methods. These groups were sometimes preparatory to the individual path, and other times were the appropriate continuation of the therapeutic work done in individual interviews.

The psychoeducational group for anxiety management was an activity organized ad hoc with a small group of four or five participants on a weekly basis for information and support. This shared virtual space, created using the Microsoft Teams platform, represented a suitable space for students to share their difficulties and confront their and their peers’ anxiety, and develop appropriate coping strategies. 

The psychoeducational intervention, with a multimethod approach, was used to: (a) provide cognitive tools of disturbances of anxiety and its physiological manifestations (emotive, cognitive, and behavioral); (b) increase awareness of coping strategies to stress-related situations, such as escape, avoidance, or hypercontrol; and (c) illustrate psychological tools and techniques aimed at modifying dysfunctional strategies and facilitating more functional methods for better coping, such as maneuvering strategies.

The workshop on study methods conducted by a psychologist or psychotherapist expert in learning psychopathology took place weekly and was addressed to a small group of three to six participants. The path followed the model for promoting academic success and enhancing study skills proposed by the MT Team of Padua (Italy) [30,31,32]. However, the explored variables in the various modules—motivation, organization, understanding, elaboration, memory, review and strategies for preparing for a test, anxiety and resilience—were adjusted in light of the new distance learning. In particular, strategies related to “taking notes in class”, “simulating the oral exam”, and “preparing for a written test” were customized in relation to exams on the Microsoft Teams platform.

Following individual paths, and after the group for anxiety or the laboratory on study methods [33], a two-month follow-up was organized as per the normal protocol of the Center. The counseling path sometimes also led to the need for further formalized and lasting help for students; the Center of Psychological Counseling has an agreement with local health structures in order to refer students for further psychiatric and/or psychotherapy help. 

## 6. Discussion

The literature on Covid-19 shows that many people have experienced psychological suffering, and psychological support has proven to be important to encourage adaptation and wider community resilience.

Here, we described the results of monitoring the psychological health of university students at a Counseling Center in Salerno, Italy, and the online services of the Center at the time of the Covid-19 phase of the lockdown.

Liu et al. [5] monitored online psychological-counseling services. They noted that they had been widely established by mental-health professionals in medical institutions, universities, and academic societies across China, and that they had facilitated the development of emergency interventions for the public and the effectiveness of emergency interventions. Online psychological interventions in times of emergency can, therefore, be fundamental to increase resilience in the population and identify any psychological problems, in order to implement timely management. This is also possible through university psychological-counselling centers, as our experience has shown. 

The *first aim* of our study was to analyze the psychological condition of the interviewed students, the components relating to study and distance learning, and some quality-of-life indicators. It emerged from our sample that anxiety and stress, concentration disorders, depressive experiences related to a sense of helplessness, and psychosomatization had been experienced by most of the interviewed students during the lockdown [34,35,36,37,38]. 

For students, the experienced symptoms were acute; in several cases, there was a reactivation of previous trauma, and an anxiety disorder became omnipresent in their lives. The prevalence of depressive experiences found in our patient sample did not emerge from PTSD from Covid-19, but only through a reactivation of previous trauma (unresolved grief, violence; 25%).

Regarding anxiety and depression, our data are in line with the literature (100%). Tang et al. [39] investigated the prevalence of and correlation between PTSD and depressive symptoms one month after the outbreak of the Covid-19 pandemic in a sample of Chinese university students in home quarantine. Extreme fear was the most significant risk factor, capable of producing psychological distress, followed by sleep problems. Yang et al. [40] highlighted how economic effects, effects on daily life, and delays in academic activities were positively associated with anxiety symptoms.

In Italy, at the time of Covid-19, about one-third of people experienced symptoms of peritraumatic stress: mild, moderate, and severe. Females had higher scores than males, and older people were more resilient than younger ones were. As for the results, the Italian participants reported similar distress values to the Chinese champion, and it was also revealed that higher stress and anxiety scores were reported by young subjects (18–50 years) [41]. According to the literature [42,43], the reactivation of previous traumas (25%) is caused by “nonintegrable” events in a person’s previous psychic system. They are capable of constituting a threat that can destructure and fragment mental cohesion. The traumatic experience can remain dissociated from the rest of the psychic experience, causing dissociative symptomatology. All students whom we observed with previous trauma showed dissociative symptoms and were sent to the territorial centers of mental health. Gritsenko et al. [44], in Russia, noted high use of painkillers, sedatives and antidepressants without a prescription in students. In our sample, we did not mention use of these substances, but for some students, psychiatric referral to the territorial mental-health services affiliated with the Center of Psychological Counseling was necessary for clinical symptoms that required psychopharmacotherapy.

From our data, psychosomatic manifestations (80%) emerged that were connected to anxiety disorders, distress, and depressants. The data are in line with the study of Liu et al. [13], which also verified the incidence of somatic symptoms correlated with depression and anxiety. Shevlin et al. [45] indicated that having high Covid-19 anxiety scores was associated with an increase in somatic symptoms. Lastly, Khan et al. [46] identified that the perception of physical symptoms from Covid-19 was significantly associated with the stress, anxiety, and depression subscale. Furthermore, fear of infection, financial uncertainty, a lack of exercise, and limited or absent recreational activities were correlated with greater symptoms of distress, anxiety, depression, and PTSD.

Our data also showed that forced social isolation co-participated in the manifestations of psychological difficulties or actual clinical symptoms. This is also in line with the literature. In this regard, Jordan et al. [47] showed that spacing measures made certain categories of people more vulnerable, risking further loneliness, isolation, and loss of mental and physical function. Pancani et al. [48] assessed the psychological repercussions of objective isolation in Italy. Results showed that the greater the isolation was, the smaller the physical space in which people were quarantined, and the worse their mental health (for example, due to the onset of depression).

Conflict with family members often contributed to this, as our data showed (53%). Pan [49] and Kant [50] conducted different studies with the same results, noting that Covid-19 offered students the opportunity to stay with their family all the time, but with different outcomes. Of the students, 88% admitted that they felt more attached to their family; however, 22% said that they had experienced difficulties due to an increase in conflict with their parents. In our sample, these difficulties emerged in a much more evident way.

With respect to distance learning, Italy promptly made up for the blocking of training and education activities with online teaching, as did most global governments. For Pragholapati [51], more than 91% of the world’s student population benefited from distance learning for the continuation of school and university education. The students that we interviewed highlighted how tiring distance learning was, and how it affected their study and study methods (51%). Internet connection problems, not being in physical proximity to teachers and peers, and being forced to stay in a closed albeit familiar environment created inconveniences. Nevertheless, the unconsolidated practice of online teaching meant that students found themselves following many hours of lessons in a manner similar to those in attendance. These data are in line with Bao’s research [52], in which students were interviewed about online study, and highlighted concerns about Internet connectivity and teachers’ preference for asynchronous- rather than synchronous-delivery teaching methods. Liu et al. [53] highlighted the difficulties related to distance learning, in particular due to the weakness of the technological infrastructure of online teaching, the inexperience of teachers, and the complex home environment.

Lastly, we examined quality-of-life indicators such as appetite, sport, sleep, use of alcohol and smoking, and in general the declaration by subjects of “feeling fit”. Our data showed that quality of sleep was very poor (73%) in most cases of the interviewed students; moreover, in general, the interviewed subjects stated that they did not feel very fit, and that they practiced very little motor and sport activity. There were no major variations in smoking or alcohol consumption during the lockdown phase. Appetite underwent only partial changes. In this regard, Wang et al. [54] compared the dietary intake of university students interviewed in 2019 and during Covid-19. They noted that total 24-h energy intake during the Covid-19 pandemic tended to be higher than that in 2019 (+13.2%). Regarding the amount of physical activity, these researchers identified a significant reduction in motor and sport activity. With respect to sleep, the studies of Simpson and Manber [55], and Cellini et al. [56] highlighted sleep disturbances likely related to increased stress/anxiety for many individuals, as well as significant behavioral changes resulting from having to stay at home. According to the cited research, the increase in sleep difficulties was more evident in people with a high level of depression, anxiety, and stress symptoms, and was often associated with the perception of time stretching [57]. This expanded temporal perception was also reported by our respondents. 

The *second aim* of our study was to describe the results of psychological-counseling support.

The follow-ups are in progress, but here they report the general observations of the psychotherapists at the various services:


*Online Counseling Interviews*


This service helped students to interpret the reality of the pandemic with greater clarity and awareness thanks to the support of expert psychologists and psychotherapists; to avoid anxiety and stressful situations due to Covid-19’s impact on the routine of daily life, which can lead to psychopathological disorders in some cases; and to ensure that the results of distance learning were positive or at least not penalizing for study and academic success for those students who had experienced it in an anxious or problematic way. 


*Online Psychoeducational Group for Anxiety-Management Interventions*


This service was a secure relational context. In fact, the students said that they felt less alone and less pathological, and, as someone declared, “more normal”, recognizing the not necessarily pathological character of anxiety. In this sense, the group had a function of containment, comparison, elaboration, and positive emotional drive.


*Online Workshop on Study Methods*


This service was related to the motivation and organization of the study. These aspects were the most negatively affected by the lockdown: almost every student reported great difficulty in organizing (and reorganizing) their time, and this is likely to have been very destabilizing. Most of them also reported a drop in motivation caused by the absence of comparison with colleagues, and the lack of continuous and immediate feedback that teachers can provide in person.

The limitations of the study are related to the size of the sample and the absence of a control group. It being an exploratory study, there are a number of systematic methodological errors. Future prospects concern comparing the results between the lockdown and subsequent phases, completing clinical follow-ups.

## 7. Conclusions and Recommendations 

The Covid-19 emergency led to a change in the habits and daily lives of most of the global population, including the academic communities of universities, which generally base a large part of their lives on interpersonal and social relationships.

During the lockdown, not all people had sufficient resources and maneuvering tools to be able to adequately manage the psychological stress, fears, and anxieties generated by the exceptional emergency event that suddenly involved the whole world. In the confinement phase, characterized by high stress and concerns deriving from the restrictions imposed by the emergency, many people found themselves experiencing for the first time more or less intense symptoms of fear, anxiety, worry, and irritability, or seeing a pre-existing anxious condition worsen.

We did not find many studies on concrete experiences of psychological support to university students at the time of the lockdown for Covid-19, especially in the Italian context; so, in our opinion, the strength of this paper is described as an online clinical experience. The effectiveness of this online model could help other facilities as a reference model.

## Figures and Tables

**Table 1 healthcare-08-00440-t001:** Area 1: psychological manifestations.

1.1 Anxiety disorders		
		Percentage
ANSWERS	No	0
	Yes	100.0
1.2 Anxiety, depression		
		Percentage
ANSWERS	No	12.2
	Yes	87.7
1.3 Negative thoughts, images, memories, and feelings (revisiting)		
		Percentage
ANSWERS	No	16.3
	Yes	77.6
1.4 Repercussions on self-esteem and self-efficacy		
		Percentage
ANSWERS	No	20.4
	Yes	79.6
1.5 Reactivation of a previous trauma (post-traumatic stress disorder (PTSD))		
		Percentage
ANSWERS	No	75.5
	Yes	24.5
1.6 PTSD from Covid-19		
		Percentage
ANSWERS	No	100
	Yes	0

Participants, 49; no missing answers.

**Table 2 healthcare-08-00440-t002:** Area 2: psychosomatization.

		Percentage
ANSWERS	No	20.4
	Yes	79.6

Participants, 49; no missing answers.

**Table 3 healthcare-08-00440-t003:** Area 3: family and social relations.

3.1 Stress due to forced cohabitation with family		
		Percentage
ANSWERS	No	46.9
	Yes	53.1
3.2 Difficulties in long-distance relationships (loss of friendships, end of romantic relationships)		
		Percentage
ANSWERS	No	16.3
	Yes	83.7
3.3 Difficulties related to social and relational limitations		
		Percentage
ANSWERS	No	30.6
	Yes	69.4

Participants, 49; no missing answers.

**Table 4 healthcare-08-00440-t004:** Area 4: distance learning/study difficulties and related matters.

4.1 Difficulties related to distance learning (online lessons, online exam anxiety)		
		Percentage
ANSWERS	No	49.0
	Yes	51.0
4.2 Difficulties in studying (methods, study organisation, poor concentration)		
		Percentage
ANSWERS	No	51.0
	Yes	49.0
4.3 Difficulties in organising time and perception of its expansion		
		Percentage
ANSWERS	No	49.0
	Yes	51.0

Participants, 49; no missing answers.

**Table 5 healthcare-08-00440-t005:** Area 5: quality-of-life indicators.

5.1 Declaration of feeling fit		
		Percentage
ANSWERS	No	65.3
	Yes	34.7
5.2 Declaration of having slept well		
		Percentage
ANSWERS	No	73.5
	Yes	26.5
5.3 Declaration of having had appetite changes		
		Percentage
ANSWERS	No	57.1
	Yes	42.9
5.4 Declaration of having practised sports		
		Percentage
ANSWERS	No	81.6
	Yes	18.4
5.5 Declaration of having smoked more		
		Percentage
ANSWERS	No	81.6
	Yes	18.4
5.6 Declaration of having consumed more alcohol		
		Percentage
ANSWERS	No	95.9
	Yes	4.1

Participants, 49; no missing answers.

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
