# Peer review of "Online University Counselling Services and Psychological Problems among Italian Students in Lockdown Due to Covid-19"

_healthcare, 2020, doi:10.3390/healthcare8040440_

Round 1
Reviewer 1 Report
The paper deals with a very important topic, and the authors have made significant improvement throughout the paper.
However, the paper has serious methodological problems. It seems that the study was poorly designed, so the authors are confronted with problems which cannot be fixed. The original version stated that 217 university students participated in an online survey and an additional 49 university students who were undergoing psychological treatment during the Covid-19 lockdown participated in the study. The revised version is based on a study of 49 students. This new sample size would have been fine for a qualitative study, but for a predominantly survey-based study, this number is statistically insignificant. The very small sample size has implications with respect to the paper’s findings. Yes, this is an exploratory study, but scientific research is guided by protocols which must be respected. The authors need to figure out how best to use data collected from the 217 university students, as this number is fine for an exploratory quantitative study.
Author Response
Dear Reviewer,
Thanks for your important feedback.
We have tried to answer at your suggestion.
We have reinserted the 217 students, but of these students we discuss only the general data, because the new version of the paper is based only on the 49 subjects who have completed the psychological counseling path

Reviewer 2 Report
In general I thought the author(s) did an good job in addressing the reviewers' comments in the revised manuscript. However, some areas still need clarification as noted below:
The title should be revised, it is suggested "Online University Counseling Services and psychological problems among italian students in lockdown due to Covid-19".
Regarding the objectives of the study, these should be changed, that is, objective 2 should be objective 1 and the opposite. It starts with the description of the University’s online psychological counseling treatment service, and only later does it make sense to analyze the psychological difficulties and mental health problems among students.
Should data collection instruments be described in more detail, were the ad-hoc questions built on the literature? What?
Some psychometric information for the SCL-90-R should be added.
The presentation of the results must be changed, according to the reorganization of the objectives.
Author Response
Dear Reviewer,
Thanks for your important feedback.
We have tried to answer all your suggestions.
We have arranged the objectives and made them coherent in the various parts of the text: abstract, objectives, introduction, results, discussion
We have specified the source of the ad hoc questions
We have added some psychometric information for SCL-90-R

Reviewer 3 Report
Comments to the Authors
I appreciate the opportunity to re-read this revised manuscript. I found that the authors have been responsive to the reviews. With some additional clarifications, this paper stands to make some contributions to the literature regarding psychological problems in lockdown due to Covid-19.
Abstract: I’m not sure I understand the wording in the following sentence: “A semi-structured interview conducted at each student’s initial intake, which assessed general mental status and general mental functioning; 2) a survey consisting of sociodemographic and ad hoc questions; and 3) the SCL-90-R test.”
Pag. 5, line 126: I’m not sure I understand the wording in the following sentence: “The SCL-90-R test [19], which investigates the possible presence of psychopathological traits.”
Pag. 5, line 210, ethical considerations: I’m not sure about this declaration: “In Italy, there is no obligation to consult an ethics committee for psychological studies.” If the authors’ problem is that an IRB is not available at the affiliations of the researchers involved in the study, I would suggest writing as follows: “The research complied with the Ethics Code of the Italian Psychology Association (Associazione Italiana di Psicologia - AIP, 2015) that draws inspiration from with WMA-Declaration of Helsinki (1964/2013). As no Institutional Review Board for Psychology research was available from the affiliations of the researchers involved in the study (i.e. University of XXX), no request for approval could be submitted.”
Pag. 5: Authors write as follows: “The results of the students’ responses to the interview and to the semi-structured questionnaire…” Do the authors intend “questionnaire” and “semi-structured interview”?
Results, pag 5: I would suggest differentiating what are the quantitative results of the students’ responses to the questionnaire and what are the qualitative responses deriving from the interviews through the Result section.
Limitation section, pag. 13, lines 539-541: This sub-section has to be included in the Discussion section (not in the Conclusion section). Furthermore, I appreciate that the authors added more to the limitations of their study, but I think it could be stated more directly about the systematic errors for methodological aspects.
Minor: There continue to be some grammatical issues throughout, including a need for consistent verb tense.
Author Response
Dear Reviewer,
Thanks for your important feedback.
We have tried to answer all your suggestions.
We did the language review at the Journal service and cleared up all doubts about the language and meaning
We have included your suggestion for ethical considerations and limitations. Thank you!
We corrected page 5

Reviewer 4 Report
The quality of the study has improved since last time.
I salute you for your hard work.
Author Response
Dear Reviewer,
Thanks for your important feedback.
We have tried to answer all your suggestions.
Thank you for your help!

Reviewer 5 Report
- There is no conclusion in Abstract.
- The study set two aims. The reasons why the study should be conducted need to be stated in more detail.
- The authors consider that there was not possible to have a comparison group. However, in limitation section, they think the future research can complete clinical follow-ups to enable a comparison of the pre-test and post- test data. This controversy may have affected the contribution of the current study.
- The semi-structured interview was provided by the Order of Italian Psychologists. The authors are suggested to describe the validity and application of the Order.
- In method section, the SCL-90-R, Sf-12 test and psychological counselling course need to be addressed more detail.
- The authors consider there is no obligation to consult an ethics committee, which needs to be made sure whether it meets the publication criteria of the Journal.
- 3.4 Measurements section is redundant.
- In the results section, there are many redundant statements. The Tables demostrated poorly. An aggregated Table including 5 areas is suggested.
- “5. Notes on Online Services” section needs to be revised more concisely. I think several statements should be moved to the Methods and Discussion section. Moreover, the students’ outcomes after the online services need to be reported and discussed. As the authors concluded, the strength of this paper is its description of an online clinical experience. The effectiveness of online model can be provided to the other facilities as a reference.
- Part of demostrations in Discussion section are not shown in Results. For example, quality of sleep was in most cases very poor among the students interviewed (73%).
Author Response
Dear Reviewer,
Thanks for your important feedback.
We have tried to answer all your suggestions.
We have added the conclusions in the abstract
We have clarified the reasons for ours aims
We have fixed the limitation section as your suggestion
We have clarified the details of semi-structured interview by the Order of Italian Psychologists
In the method section,
we have included specifications for the SCL-90-R test. We do not mention the Sf-12 test anymore
We have clarified the ethical considerations better
We have eliminated the Measurements section
In the results,
we have placed many redundant statements
We have made the section on notes on Online Services more concise
We have entered the effectiveness of online model for clinical intervention
In the Discussion section, we have entered the result of the quality of sleep

This manuscript is a resubmission of an earlier submission. The following is a list of the peer review reports and author responses from that submission.
Round 1
Reviewer 1 Report
TITLE: Psychological reactions of Italian university students to lockdown from Covid-19 and taking charge at a University Counselling Centre. A multimethod study
The present study aims at investigating a relevant area of research. However, several limitations have to be addressed both in the structure of the article including the introduction where previous studies are uncorrectedly cited. Furthermore, a clear description of the instruments adopted for psychiatric diagnosis should be reported improving the method section and the clearness of the results reported. In particular, specific revisions should address the following points:
1) The references cited in the text are without order and need to be fixed
2) The introduction must be more organized and related to the major aims.
2) The study methods are unclear and incomplete. It is not clear how the students were recruited. Furthermore, it is not clear how the intensity of the psychopathological symptoms investigated in the 5 areas of the questionnaire was assessed. No clear description of the instruments adopted for psychiatric diagnosis is reported.
3) In the results it emerges a percentage of University students had a recall of trauma experienced in the past but none of them have developed a PTSD linked to Covid. Is this a full or partial diagnosis and how do you assess it?
4) Discussion is too loose and should be reorganized and resynthesized
Author Response
Dear Reviewer,
Thank you for your important feedback. We have eliminated the first part of the study (using online data) and have reported only the aims, methods and results of the experience of the Counselling Centre. We made this decision because the various reviewers helped us understand that there was a great deal of confusion in the previous version. Furthermore, we believe that focusing the paper on clinical work is much more original.
We have attempted to respond to your suggestions as follows:
- We have cited the references in the text in order.
- We have organised the introduction and related it to the major aims.
- We have clarified the study methods, instruments and procedures.
- We have clarified in the results the diagnosis of PTSD.
- We have reorganised and re-synthesised the discussion.
Reviewer 2 Report
I think the structure of the article including introduction, method, results, and discussion is too loose to be read concisely. The references cited in the main text are orderless. There are several major flaws in study methods. For example, the authors did not describe how these participants were chosen. Why did they think the participants can represent the whole college students? Why such a sample size can afford to test the significance of the difference between two groups. How can they indicate the level of anxiety and stress of the participants is high? Is there any control/compared group? Instead of multiple variable analysis, the Chi-square test for one factor is not suitable to be applied to social science or psychological study. “anxiety and anguish (p = 0.5)” should not be significant. Moreover, for ethical consideration, a formal IRB approval should be warranted.
Author Response
Dear Reviewer,
Thank you for your important feedback. We have eliminated the first part of the study (using online data) and have reported only the aims, methods and results of the experience of the Counselling Centre. We made this decision because the various reviewers helped us understand that there was a great deal of confusion in the previous version. Furthermore, we believe that focusing the paper on clinical work is much more original.
We have attempted to respond to your suggestions as follows:
- We have cited the references in the text in order.
- We have organised the introduction and related it to the major aims.
- We have clarified the ethical considerations, study methods, participants, instruments and procedures.
- We have clarified that this is an exploratory study and eliminated the crosstabs and chi-squared results.
- We have reorganised and re-synthesised the introduction and discussion.
Reviewer 3 Report
I appreciate the opportunity to review the "Psychological reactions of Italian university students to lockdown from Covid-19 and taking charge at a University Counselling Centre. A multimethod study". As an educator, I read this study very interestingly. Thank you for the authors' efforts.
[1] Introduction
- I hope the authors highlight the reason why this study should proceed.
[2] Conclusion
- The authors should demonstrate the implications and strengths of the research that can be derived from the results of this study.
[3] limitations of research
- There is no mention of the limitations of this study and the direction of future research.
Author Response
Dear Reviewer,
Thank you for your important feedback. We have eliminated the first part of the study (using online data) and have reported only the aims, methods and results of the experience of the Counselling Centre. We made this decision because the various reviewers helped us understand that there was a great deal of confusion in the previous version. Furthermore, we believe that focusing the paper on clinical work is much more original.
We have attempted to respond to your suggestions as follows:
- We have highlighted the reasons why this study should proceed.
- We have discussed the limitations of the study and suggested directions for future research.
- We have reorganised the discussion and have demonstrated the implications and strengths of the research that can be derived from the results of this study.
Reviewer 4 Report
Healthcare Journal
September 2, 2020
Subject: review of the article titled: “Psychological reactions of Italian university students to lockdown from Covid-19 and taking charge at a University Counselling Centre. A multimethod study” (ID: healthcare-907207)
Review Report
I really appreciated the opportunity to read this interesting paper that explore the mental health of Italian university students during the Covid-19 pandemic. The paper is interesting, relevant and quite original. The study would be of interest to the Healthcare Journal readers. I think more specificities are needed in the Method, Results and Discussion Sections.
Specific comments:
- Title: I would suggest a more concise and effective title.
- Introduction section (lines 29-32): This section is well written and complete. The literature revision is appropriate and extensive. I would suggest to describe some of the main characteristics of the Italian lockdown. The authors have already reported the specificities in the Discussion section (lines 376-400), but I would suggest to anticipate some sentences in order to introduce the readers to the Italian situation.
Lines 55-62: As for the Salman et al’s study, I do not understand the “acceptance” coping strategy.
- Aims (lines 150-156): Did the authors expected (specific) mental health consequences about the lockdown situation? If yes, it would be interesting to understand if it is expected that the COVID 19 lockdown pandemic is the main factor triggering mental problems or if it is the pandemic itself (beyond the lockdown) to affect health.
- Participants (Line 158): What does it mean out-of office students? The authors also used the terms current student/out-of-course student, on-site student/off-site student. I do not understand well these terms and the differences. I would suggest to better explain this point. Then, I would suggest to modify the term “Average” with the italics symbol “M”, and to modify the term “Standard deviation” with the symbol “SD”. Finally, the authors anticipated a result in this Participants section: the five students who were sent to territorial mental health units for psychiatric consultations (line 170). I would suggest to delete this sentence from this section.
- Line 172: I would suggest to entitled the Section “Measures” or “Materials” deleting the terms “data collection”. I would suggest to better describe the instruments that the authors used. For instance, is the Sf-12 Test an acronym for a validated instrument? If yes, who are the authors? Is it composed by dimensions? Could the authors added some examples of items? More specificities are also needed for the interview conducted (e.g., what typology of interview?)
- Results:
- Tables: If items are presented on tables it would be good to report the number of participants and missing data for each measure.
- Chi square test (lines 369-374): I would suggest to better describe the results of Chi-square test and the relating indexes.
7)Discussion: I would suggest to insert the study limitations. Finally, the paragraph 6 (lines 492-568) should be in the Result section, I suppose.
Author Response
Dear Reviewer,
Thank you for your important feedback. We have eliminated the first part of the study (using online data) and have reported only the aims, methods and results of the experience of the Counselling Centre. We made this decision because the various reviewers helped us understand that there was a great deal of confusion in the previous version. Furthermore, we believe that focusing the paper on clinical work is much more original.
We have attempted to respond to your suggestions as follows:
- Title: We have made the title more concise.
- Introduction section (lines 29–32):
- We have described the main characteristics of the Italian lockdown.
- We anticipate in the introduction the specificities in the discussion section (lines 376–400).
- We have clarified Salman et al.’s study’s discussion of the ‘acceptance’ coping strategy (lines 55–62).
- We have clarified the aims.
- Participants:
- We have clarified the terms current student vs. out-of-course student and on-site student vs. off-site student.
- We have replaced the term ‘Average’ with the italicised symbol ‘M’ and the term ‘Standard deviation’ with the symbol ‘SD’.
- Regarding the phrase ‘the five students who were sent to territorial mental health units’, we have deleted this sentence from the section on psychiatric consultations. We have inserted it in the results section.
- We have titled the section ‘Measures’, deleting the term ‘Data collection’.
- We have described the instruments more accurately.
- Results:
- We have reported the number of participants and missing data for each table.
- We have clarified that this is an exploratory study and eliminated the crosstabs and chi-squared results.
- Discussion:
- We have discussed the limitations of the study and suggested directions for future research.
- We have inserted paragraph 6 (lines 492–568) in the results section.
Reviewer 5 Report
The literature addressed is described accurately so far as I can see. The method seems to have been followed faithfully and the authors were well-positioned to conduct the analysis. The material is interesting and the topic is relevant. Despite these positives in my view the paper needs more work before it could be published and I have made some specific suggestions below.
Three main aim are presented, however the aim 1 as it is presented, seems more of a purpose of the study. I suggest your review.
Method section
- The methods section should be explained in more detail. The method section would benefit, from being presented in two interrelated studies. In study 1 the characterization of the psychological health of Italian university students during the Covid-19 lockdown. And in study 2 the application of objectives 2 and 3.
- The data collection should be explained in more detail. When were the data collected? Data collection period? Response rate? How were participants selected? Any criteria for selection? There is no mention of the sample size that was targeted and obtained to meet the sample size requirements for data analysis. The data were normally distributed? This information should be provided.
- The SF-12 instrument reference must be added. The items of both questionnaires were administered in English only or in Italian? If so, this probably means that the instruments were adapted by the authors of the present article. What criteria were used in the translations and cultural adaptations?
- Tables should be numbered from 1.
- Please clarify how the questionnaire for clinical care at the Centre of Psychological Counselling was chosen/developed. Was it based on a valid/reliable previous instrument? Was it developed specifically for the study? If the last, how was it developed?
- The authors report that “five students were sent to territorial mental health units for psychiatric consultations”. What behavioral/cognitive changes were found that justified such a decision (e.g. suicidal ideation, self-injurious behaviors).
Data Analysis
- There is little explanation as to why you analyse your data in the way that you do, or why your methods/statistical procedures are appropriate. This should be clarified.
Discussion section
- There is a complete absence of the empirical implications of the study, besides which the theoretical implications should have been approached in greater depth. Identify recommendations for practice/research/education as appropriate, and consistent with limitations.
CHECKLIST FOR STYLE
- The manuscript will serve a broad audience of students, researchers, and practitioners, however the manuscript needs to be carefully and attentively proofread by an English-speaker, because many sentences are awkwardly constructed, punctuation and therefore the reading is occasionally difficult to follow.
Author Response
Dear Reviewer,
Thank you for your important feedback. We have eliminated the first part of the study (using online data) and have reported only the aims, methods and results of the experience of the Counselling Centre. We made this decision because the various reviewers helped us understand that there was a great deal of confusion in the previous version. Furthermore, we believe that focusing the paper on clinical work is much more original.
We have attempted to respond to your suggestions as follows:
- We have explained the instruments, procedures, participants and data collection in more detail.
- We have numbered the tables beginning at 1.
- We have clarified the questionnaire for clinical care at the Psychological Counselling Centre
- We have clarified the behaviour of the ‘five students were sent to territorial mental health units for psychiatric consultations’.
- We have clarified that this is an exploratory study and provided an explanation of the analysed data.
- We have discussed the limitations of this study and suggested directions for future research.
- We have had the paper proofread by an English speaker.
- We have identified recommendations for practice, research and education as appropriate.
Reviewer 6 Report
The methodology section has serious flaws which must be addressed before the paper can be considered publishable. First, what is the sampling technique used in the study? Second, why was that sampling technique selected? Third, how did the authors deal with the issue of bias? Fourth, how representative is the sample? Put differently, can we say that the sample is representative of Italian university students? The point I’m emphasizing here is that the authors must demonstrate evidence of methodological rigour, which is vital, especially in quantitative studies.
Author Response
Dear Reviewer,
Thank you for your important feedback. We have eliminated the first part of the study (using online data) and have reported only the aims, methods and results of the experience of the Counselling Centre. We made this decision because the various reviewers helped us understand that there was a great deal of confusion in the previous version. Furthermore, we believe that focusing the paper on clinical work is much more original.
We have attempted to respond to your suggestions as follows:
- We have clarified that is an exploratory study and eliminated the crosstabs and chi-squared results.
- We have reported the number of participants and missing data for each table.
- We have reorganised the discussion and demonstrated the strengths of the research and implications that can be derived from the results of this study.
- We have discussed the limitations of this study and suggested directions for future research.